# Double-filtered leukoreduction as a method for risk reduction of transfusion-associated graft-versus-host disease

Sejong Chun[1], Minh-Trang Thi Phan[2], Saetbyul Hong[3], Jehoon Yang[3], Yeup Yoon[3,4], Sangbin Han[5], Jungwon Kang[6], Mark H. Yazer[7], Jaehyun Kim[6]*, Duck Cho[2,4]*

1 Department of Laboratory Medicine, Chonnam National University Medical School & Hospital, Gwangju, Korea, 2 Department of Laboratory Medicine and Genetics, Samsung Medical Center, Sungkyunwan University School of Medicine, Seoul, Korea, 3 Animal Research and Molecular Imaging Center, Samsung Medical Center, Seoul, Korea, 4 Samsung Advanced Institute for Health Sciences & Technology, Sungkyunwan University School of Medicine, Seoul, Korea, 5 Department of Anesthesiology and Pain Medicine, Samsung Medical Center, Sungkyunkwan University School of Medicine, Seoul, Korea, 6 Blood Transfusion Research Institute, Korean Red Cross, Wonju, Korea, 7 Department of Pathology, University of Pittsburgh, Pittsburgh, PA, United States of America

* duck.cho@skku.edu (DC); kimjh@redcross.or.kr (JK)

**Data Availability Statement:** All relevant data are within the paper and its Supporting Information files.

## Abstract

### Background

Transfusion-associated graft-versus-host disease (TA-GvHD) is caused by leukocytes, specifically T cells within a transfused blood product. Currently, the prevention of transfusion-associated graft-versus-host disease is performed by irradiation of blood products. With a sufficient reduction of leukocytes, the risk for TA-GvHD can be decreased. With consistent advances in current state-of-the-art blood filters, we herein propose that double filtration can sufficiently reduce leukocytes to reduce the risk for TA-GvHD.

### Materials

Thirty RBC concentrates were filtered with leukocyte filters, followed by storage at 1–6 °C for 72 hours, and then a second filtration was performed. Residual leukocytes in the double-filtered RBC units (n = 30) were assessed with flow cytometric methods, and an additional assay with isolated peripheral blood mononuclear cells (PBMCs) (n = 6) was done by both flow cytometric methods and an automated hematology analyzer. Quality of the RBCs after filtration was evaluated by hematological and biochemical tests. *In vitro* T cell expansion was performed using anti-CD3/CD28-coated Dynabeads or anti-CD3 (OKT3). *In vivo* experiment for GvHD was performed by using NOD.Cg-Prkdc[scid] Il2rg[tm1Wjl]/SzJ (NSG) mice.

### Results

Double-filtered blood products showed residual leukocyte levels below detection limits, which calculated to be below 1200–2500 cells per blood unit. *In vitro* expansion rate of T cells showed that $6\times10^3$ and $1\times10^3$ cell-seeded specimens showed 60.8±10.6 fold and 10.2±9.7-fold expansion, respectively. Cell expansion was not sufficiently observed in wells

**Funding:** This study was supported by a grant provided by the Blood Transfusion Research Institute, Korean Red Cross (2016-I02), as an inside research project, and in part by Basic Science Research Program through the National Research Foundation of Korea (NRF) funded by the Ministry of Education, Science and Technology (2015R1D1A1A09058740 and 2018R1A2B6006200). The funders had no role in study design, data collection and analysis, decision to publish, or preparation of the manuscript.

**Competing interests:** The authors have declared that no competing interests exist.

planted with $1 \times 10^2$ or 10 cells. *In vivo* experiments showed that mice injected with $1 \times 10^5$ or more cells cause fatal GvHD. GvHD induced inflammation was observed in mice injected with $1 \times 10^4$ or more cells. No evidence of GvHD was found in mice injected with $10^3$ cells.

## Conclusions

Our study suggests that additional removal of contaminating lymphocytes by a second leukodepletion step may further reduce the risk for TA-GvHD.

## Introduction

Transfusion-associated graft-versus-host disease (TA-GvHD) occurs from the engraftment of viable donor leukocytes, particularly T cells, following a transfusion of cellular products. The recipient's immune response to these infused cells would usually prevent TA-GvHD, yet in cases where the recipient is immunosuppressed, or in a situation of HLA haploidentical match between donor and recipient, the infused lymphocytes can proliferate *in vivo* and attack the host tissues[1]. Removal of these cells before transfusion has been proposed to reduce the incidence of TA-GvHD, but reports of TA-GvHD after transfusion of leukoreduced blood components can be found in the literature[2–4]. Despite these reports being from many years ago, leukoreduction by filtration is generally not accepted as a method to sufficiently to prevent TA-GvHD [5, 6] Irradiation disables T cells' mitotic capabilities and is currently the *de facto* golden standard method for full prevention of TA-GvHD, despite some incidences of TA-GvHD being reported after irradiation [7].

However, the procurement and maintenance of irradiators in hospital blood banks is a challenging task, and the cost of irradiators are increasing. Therefore, relying on blood centers or other transfusion services to perform the irradiation is a common practice, and this, in turn, can result in delays in the administration of blood products to patients and adverse effects related to the irradiation of blood components, such as hyperkalemia, is another undesirable side effect[8–11]. A study conducted in Japan reported that patients demonstrating electrocardiogram changes caused by hyperkalemia following the infusion of irradiated units have increased since irradiated blood was introduced nationwide[12].

An investigation on whether a method is suitable to fully prevent whether it can fully prevent TA-GvHD is not possible; even the current method of gamma-ray irradiation has not been subject to any randomized clinical trial on this matter[7]. We relied on investigating the level of leukocytes left after two rounds of filtration, and compared that with data on the level of mononuclear cells required to show proliferation *in vitro*, or GvHD symtomes *in vivo*. This approach was done as the underlying pathophysiology of GvHD is T cell proliferation and expansion, which requires a critical number of T cells in a blood product[13]. Under the presumption that lowering the number of residual leukocytes can significantly reduce the possibility of T cells expanding, we hypothesized that T cell expansion and proliferation might be correlated with the number of residual leukocytes. We also investigated the quality and number of leukocytes in extensively filtered blood products. With these series of experiments, we demonstrate that sufficiently filtered blood products can decrease the risk of TA-GvHD, a notion that is worth further investigation with modern blood filters. These results suggest that the double filtration, which is a simple method of an additional bedside or lab side filtration of pre-storage leucodepleted blood products, can decrease the risk of TA-GvHD, without the potential side effects and possible burdens accompanied by blood irradiation.

## Material and methods

### Evaluation of residual leukocyte levels and blood product quality

RBC units from donors with elevated ALT levels ($\geq$101 IU/dL) were selected for this study. These units are not acceptable for transfusion in Korea and are subject to discard. Red blood cells are prepared from 400 mL CPDA-1 whole blood by centrifugal separation of the cells from plasma. Thirty RBC units were selected and filtered at room temperature with RCM1 leukocyte filters (Haemonetics, Braintree, MA, USA) according to the manufacturer's directions, followed by storage at 1–6 $^o$C for 72 hours. After storage, the units were filtered using a second filter (RC High-Efficiency Leukocyte Removal Filter with Attached Set for Blood Transfusion, manufactured by Haemonetics). This process was intended to simulate the distribution of a blood product: an initial filtration by the blood supplier, a period of being held in inventory at local hospital blood banks, and a second filtration step at the bedside or hospital blood bank. The selected blood filters were used due to the ease of availability, as they are the most commonly used in Korea, which would reflect real clinical conditions. The double-filtered RBC units were placed in stationary storage in a refrigerator between 1–6 $^o$C for an additional 29 to 32 days (samples filtered once and twice, respectively. This was done to make the length of storage 35 days in total). Residual leukocytes were counted with LeucoCOUNT reagent (BD Bioscience, San Jose, CA, USA) using a FACSCanto II (BD Bioscience) flow cytometry device. The limit of detection was enhanced from the manufacturer's suggested 0.05 to 0.01WBC/µl by multiplying the amount of sample with an equally increased amount of reagent by five. The scheme of this experiment is displayed in Fig 1.

Additional methods for the precise evaluation of residual leukocytes were done using 100 mL from each of the double-filtered RBCs units (n = 6). Peripheral blood mononuclear cells (PBMCs) using Ficoll-Hypaque (1.077 g/mL density gradient; GE Healthcare Bio-Sciences AB, Uppsala, Sweden) were isolated with sufficient margin to minimalize cell loss. Harvested cells were washed twice in phosphate-buffered saline (PBS) and suspended in 400 µL of PBS. The leukocyte count in the resuspended samples was counted by Sysmex XN-2000 (Sysmex, Kobe, Japan) in the body fluid mode, according to the manufacturer's instructions and previous study results[14, 15]. The limit of detection was 1.2 cells/µL and the limit of quantification was estimated to be 5.8 cells/µL. Cell count per bag was calculated using the following formula: number of cells/µL counted by CBC machine x the resuspended volume (400 µL) x 2 (from a bag containing 200 mL, 100 mL were used for this experiment).

The number of RBCs in each unit after each filtration step was evaluated through measurements of volume and hematocrit levels before and after filtration. The post-leukoreduction RBC retrieval rate was calculated as percentages before and after filtration. Serum potassium and hemoglobin levels in the RBC units were measured before, directly after each filtration step, and again 35 days after the initial filtration to evaluate the rate of RBC hemolysis. These parameters were compared to conventional RBC units of equivalent age, which had been filtered one time with RCM1 leukocyte filters (Haemonetics, Braintree, MA, USA) before storage in a refrigerator.

### *In vitro* expandability of T cells

Samples were acquired from the discarded blood in the disposable kits following the collection of apheresis platelets. PBMCs were isolated by Ficoll-Paque 1.077 g/mL density gradient centrifugation. Isolated PBMCs were cultured for seven days in 96-well U-bottom plates (with various numbers 6 x $10^3$,1 x $10^3$, 1 x $10^2$, or 10 cells in 200 µL/well), using supplemented RPMI-1640 medium, and incubated at 37˚C in a humidified, 5% $CO_2$ atmosphere, with anti-CD3/CD28 coated Dynabeads (Invitrogen, Carlsbad, California, USA) at a bead-to-cell ratio of 1:1, according to the manufacturer's instructions. Another batch of PBMCs was exposed to 30 ng/

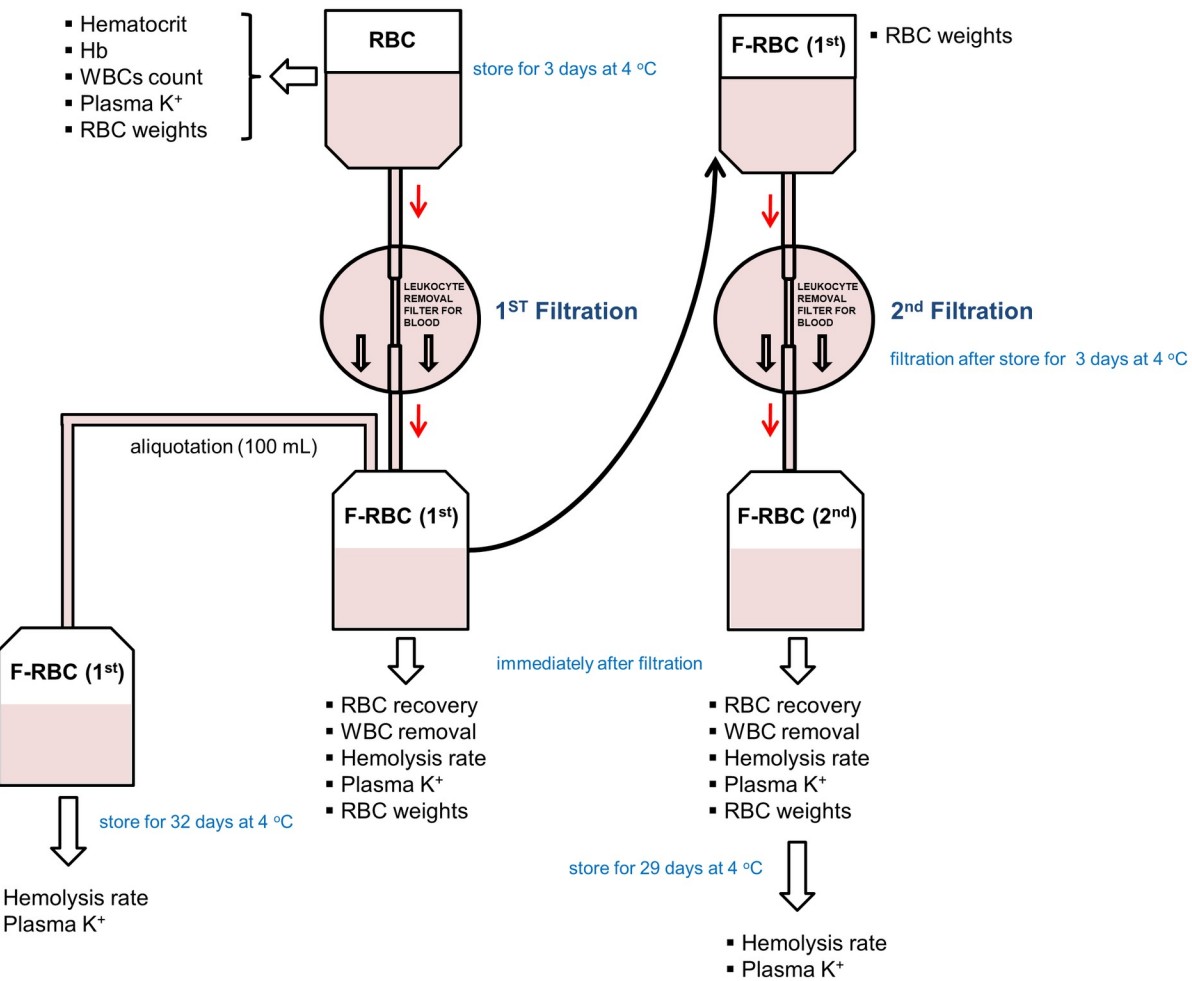

**Fig 1. Schema of the double filtration process.** 30 units of RBC products were filtered in this process. Red blood cells are prepared from 400 mL CPDA-1 whole blood by centrifugal separation of the cells from plasma. Thirty RBC units were selected and filtered at room temperature with an initial filtration process, followed by storage at 4˚C for 72 hours. Single filtrated samples were stored at 4˚C for 32 days, and samples filtered a second round were stored for 29 days at the same conditions. Parameters described in the figure was measured in each sample.

mL of OKT3 in complete RPMI 1640 medium (RPMI 1640 supplemented with 10% heat-inactivated fetal bovine serum, 100 unit/mL penicillin, 100 mg/mL streptomycin). The medium was replaced every 2–3 days with fresh medium containing 50 U/mL IL-2. Evaluation of expanded T cells was performed by cell counting using trypan blue exclusion, and the percentage of CD3+CD56-cells was determined by flow cytometry using FITC-conjugated antihuman CD3 Ab and PE-Cy5-conjugated antihuman CD56 Ab (BD Pharmingen, San Diego, CA, USA). Cell population growth was observed with a microscope.

### *In vivo* GvHD mouse model

All animal experiments followed the guidelines of the Samsung Medical Center for animal care and use, which has been accredited by the Association for Assessment and Accreditation of Laboratory Animal Care (AAALAC) International; the protocol was approved by the

Samsung Medical Center (Seoul, Korea), Animal Care and Use Committee (ACUC). In parallel to the *in vitro* T cell expansion, we induced TA-GvHD in NOD.Cg-Prkdc<sup>scid</sup> Il2rg<sup>tm1Wjl</sup>/SzJ (NSG) mice injected intravenously with various concentrations of PBMCs. Protocols were adapted from a previous study[16]. The isolation of PBMCs was performed following the protocol described above in the *in vitro* assay. Eighteen male NSG mice at eight weeks of age (bred in-house at Samsung Medical Center) were included in the study. Mice were kept initially according to groups segregated by the amount of injected PBMC (described below), and were later housed in individually ventilated cages after day 41 when mice injected with $10^4$ cells showed aggression among each other. Cages were supplied with autoclaved wood chips for bedding, and irradiated diets and sterile water for feeding. The cages were maintained at a constant temperature ($23 \pm 2$°C) and humidity ($50 \pm 10\%$) under 12 h light/dark cycle, with lights off at 20:00. Before PBMCs injection, total body irradiation (TBI) was applied to groups of three mice each by using IBL 437C (CIS Bio International, France) at a dose of 2 Gy/minute on the day before injection. After irradiation, three mice were randomly selected and housed per cage under filter tops in pathogen-free rooms and supplied with standard laboratory diet and water *ad libitum*. PBMC injection was performed in quantities of $10^7$, $10^6$, $10^5$ cells, $10^4$ cells, and $10^3$ cells. Cells were mixed with 200 μL of sterile saline before infusion. Three mice were assigned to each group, and a control group of three mice received an injection of 200 μL of saline only. The body weight of the mice was measured daily. Mice were sacrificed when their weight was below 80% of the initial measurement or when mice succumbed to GvHD. All mice were euthanized after experimentation, at day 43 by $CO_2$ gas. Liver, lung, kidney, skin, and bone marrow of all mice were histologically analyzed using hematoxylin and eosin staining. Evaluation of inflammatory cell infiltration, tissue necrosis, fibrosis, and hemorrhage was performed. Immunohistochemistry staining with anti-human CD3 antibody (clone EP449E, Abcam, Cambridge, UK) and CD45 (clone EP322Y, Abcam) was performed using the same tissues to evaluate human leukocyte infiltration.

### Ethics approval and consent to participate

The use of these blood products was approved by the institutional review board of the Korean Red Cross (assignment no. 2016-I02). Consent of the blood donor was waived for this study. The protocol employed in this study is in compliance with the Helsinki Declaration.

## Results

### Residual WBCs are lower than detectable levels in double-filtration blood products

A single filtration step resulted in 5 log reduction of leukocytes [ranging from 7796.40 cells/μL (pre-filtration) to 0.08 WBCs/μL (single filtration) and below detection levels (double filtration)]. To further verify the number of remaining leukocytes in double-filtered samples that showed below detection levels, leukocytes in samples concentrated by PBMCs isolation protocol were used for isolation of leukocytes, which was then measured by the hematology analyzer. As the limit of detection was 1.2 cells/μL[14, 15], our results showed that a double filtrated RBC unit would contain less than $10^3$ cells, when counting mononuclear cells only. When accounting for the loss during the Ficoll-Hypaque washing steps (recovery rate $60 \pm 20\%$, as suggested by the manufacturer)[17], there would be a theoretical 1200–2500 residual mononuclear cells after two rounds of filtration.

Increased hemolysis was not observed when comparing single filtered and double filtered blood products. Plasma hemoglobin and potassium slightly increased after the first filtration,

**Table 1. Characteristics of the RBC units (n = 30) after each phase of filtration.**

| Parameters | pre-filtration | post-first leukoreduction | post-second leukoreduction |
|---|---|---|---|
| weight (g) | 241.5±12.0 | 232.0±14.4 | 190.2±15.9 |
| volume (mL) | 254.3±12.6 | 244.3±15.2 | 200.2±16.8 |
| hematocrit (%) | 59.4±2.6 | 54.9±4.6 | 52.7±2.3 |
| RBC (x$10^6$/μL) | 6.7±0.3 | 6.1±0.5 | 5.8±0.3 |
| WBC (cells/μL) | 7796.40±2203.13 | 0.08±0.07 | Below detection limit |
| plasma hemoglobin (g/dL) | 0.27±0.22 | 0.44±0.32 | 0.50±0.36 |
| total hemoglobin (g/dL) | 19.9±1.1 | 18.2±1.6 | 17.5±0.8 |
| plasma potassium (mmol/L) | 7.8±2.5 | 11.3±1.9 | 11.1±2.1 |

Numeric values are given as mean ± standard deviation

but the change in these parameters was smaller after the second round of filtration (Table 1). A study done by Reverberi et al. [18] shows that irradiation results in an increase of leaked potassium levels after 7 days, and is more than twice the levels of non-irradiated RBCs. This is in contrast with this study's results, suggesting that filtration itself is less damaging compared to irradiation. Furthermore, the rate of hemolysis and level of plasma potassium at day 35 were unchanged compared to measurements from post-filtration, suggesting that two rounds of filtration did not have a detrimental compound effect on the RBCs in terms of spontaneous hemolysis or RBC integrity. The most notable difference between single and double filtration was the RBC recovery rate (Table 2), which showed that the level of RBC recovery was lower following the second round of filtration. Overall, the loss of RBCs after the second round of filtration was 25.8% compared to pre-filtration, or 16.0% compared to single filtration units. The total Hb of single filtrated RBCs is 44.5 ± 0.2 g/unit and 35.0 ± 0.1 g/unit for double filtrated RBCs. This could be problematic when applying double filtration to patients requiring multiple units of blood transfusion, as the transfused amount of RBCs would be less compared to unfiltered or single filtered units, ultimately resulting in the use of more blood unit numbers.

**Table 2. Comparison of single leukoreduced RBC units with double filtrated RBC units.**

| Parameters | single filtration (n = 30) | double filtration (n = 30) |
|---|---|---|
| RBC recovery rate (%)* | 88.3±2.4 | 74.2±4.2 |
| Total hemoglobin *(g/unit) | 44.5 ± 0.2 | 35.0 ± 0.1 |
| Residual WBCs (x$10^4$/unit) | 2.9±2.5 | 0** |
| WBC removal rate | 99.998943 | 100** |
| Hemolysis rate (%) | | |
| pre-filtration | 0.40±0.22 | 0.77±0.66 |
| post-filtration | 0.68±0.50 | 0.93±0.70 |
| Storage day 35 | 3.75±1.55 | 3.33±1.43 |
| Plasma potassium (mmol/L) | | |
| pre-filtration | 7.8±2.5 | 11.3±1.9 |
| post-filtration | 6.1±1.9 | 11.1±2.1 |
| day 35 after filtration | 38.5±3.4 | 37.7±3.0 |

Numeric values are given as mean ± standard deviation

* RBC recovery rate is calculated to the rate of the number of RBCs in the post filtered units compared to in the pre-filtered RBC units

** All 30 samples showed no residual WBCs; hence no standard deviation is shown

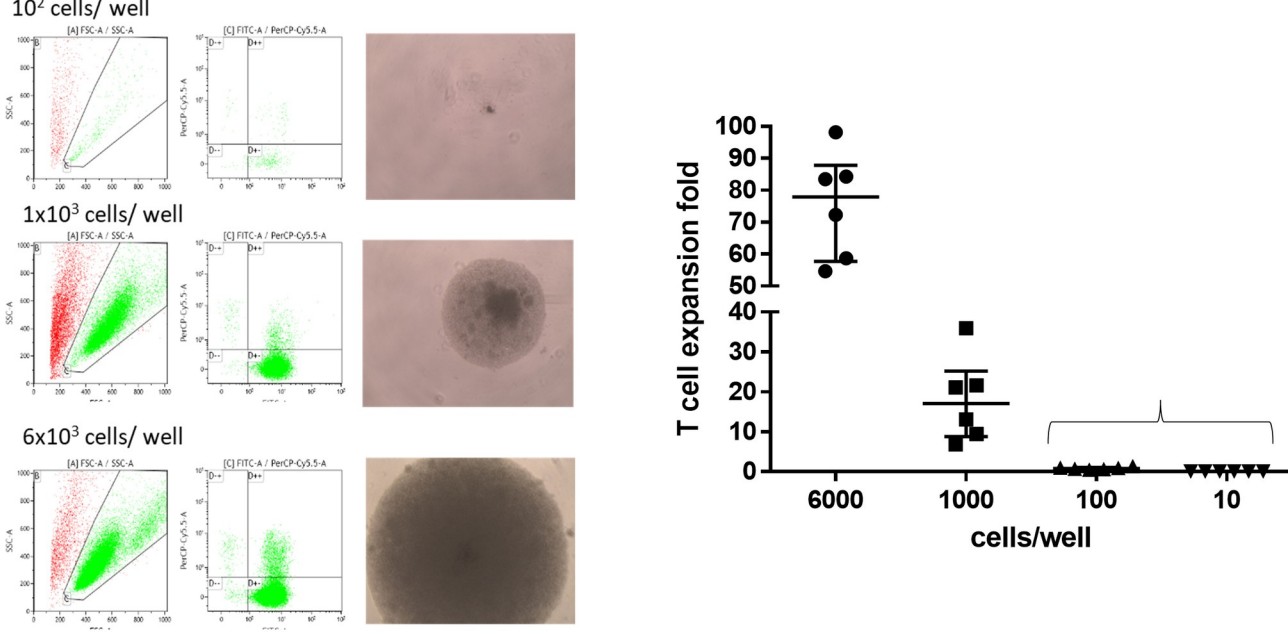

**Fig 2. *In vitro* activation and expandability of isolated peripheral blood mononuclear cells in various cell concentrations.** Cell expansion was observed in wells planted with $1 \times 10^3$ cells or more, while wells with 100 and 10 cells failed to show evidence of T cell expansion and proliferation. The initiation of proliferation can be concluded to begin when cells between $10^2$ and $10^3$ cells are planted. Sufficient proliferation was observed when $10^3$ cells are planted.

### *In vitro* T cell expansion suggests more than 1000 cells cultured in proximity for sufficient expansion

The expansion rate of T cells was evaluated after seven days of culture. Initial visual inspection indicated that wells planted with $6 \times 10^3$ cells showed sufficient expansion. Calculated results indicated that seeding of $6 \times 10^3$ or $1 \times 10^3$ cells per well resulted in a high rate of expansion of the T cells following stimulation with anti-CD3/28-coated beads (75.2 ± 16.6 fold and 18 ± 10.6 fold) or anti-CD3 (OKT3) (60.8 ± 10.6 fold and 10.2 ± 9.7 fold). However, T cell expansion was not observed for PBMCs cultured in concentrations of 100 or 10 cells/well (Fig 2). Whether a 10 to 20-fold expansion is sufficient to be interpreted as expansion and proliferation is questionable, and we interpret this results that a sufficient rate of expansion can be said to be reached when $1 \times 10^3$ or more cells are planted in proximity. The initiation of expansion would be when cells are planted in numbers between $10^2$ and $10^3$ cells. This result is similar to a report by Pohler et al.[19] that the mean frequency of proliferating T- cells in PBMCs was calculated from one in 129 plated cells. However, our result was in contrast with previous studies [20, 21] indicated that low numbers of T cells (i.e., 1 in 5 T cells) plated *in vitro* were able to proliferate. The difference can come from the methodology, as these groups stimulate T cell proliferation by co-culturing with irradiated allogeneic feeder cells, IL-2, mitogen (phytohemagglutinin), and T cell growth factor or plate-bound anti-CD3/CD28 (anti-T cell receptor) antibodies, whereas we stimulate T cells with anti-CD3/CD28 coated Dynabeads or anti-CD3 (OKT3) and IL-2, but without feeder cells or mitogen. The discrepancy among methods will require further investigation to determine the exact amount of T cells required for proliferation, as *in vitro* experimentation and actual clinical conditions may differ. It would be difficult

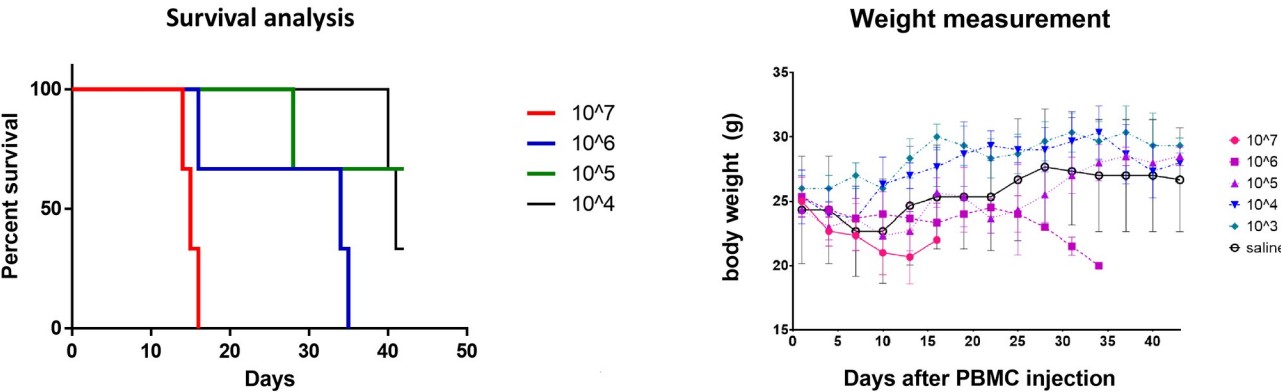

**Fig 3. *In vivo* validation on NOD.Cg-Prkdc^scid Il2rg^tm1Wjl/SzJ mice injected with various levels of human peripheral mononuclear cells.** Each group had three mice assigned to experimentation. Graft-versus-host disease (GvHD) related death was observed in mice injected with $10^5$ or more cells, while mice injected with $10^4$ or fewer cells did not show survival disadvantages due to GvHD (left). Mice injected with $10^6$ or more cells all have shown a failure to gain weight. A single mouse injected with $10^5$ cells expired at day 28, while the remaining 2 mice showed constant weight gain (right). Range in between upper and lower bars indicate 95% confidence interval range, when applicable.

to evaluate whether the 1200–2500 cells left after two rounds of leukoreduction would reach these levels of concentration in a physically proximal environment. Furthermore, *in vitro* expansion itself is not equal to the induction of GvHD, and thus further investigation with an *in vivo* GvHD mouse model was done.

### *In vivo* GvHD mouse models suggest that more than $10^4$ cells are necessary to induce inflammation

Tissue samples were obtained from the liver, lung, kidney, skin, and bone marrow of the mice that were injected with allogeneic PBMCs. GvHD-related death was observed in all mice that were injected with $10^6$ or more cells. In addition, one mouse injected with $10^5$ cells died with GvHD-related symptoms on day 28 (Fig 3, Lt). The remaining two mice injected with $10^5$ cells, and all mice injected with $10^4$ or fewer cells did not show significant weight loss or clinical features related to GvHD (Fig 3, Rt). We note that mice injected with $10^4$ cells showed aggression among each other, and two were killed overnight (day 41); however, both mice did not show symptoms related to TA-GvHD nor did they show apparent weight loss and would have been euthanized only two days later. Histological analysis revealed no inflammation in the samples from the mice that were injected with $10^3$ cells. Control mice injected with saline also showed no evidence of inflammation, as expected. The single mouse left injected with $10^4$ cells showed inflammation of the kidney parenchyma without evidence of human leukocyte infiltration in H&E or IHC stains. It is unlikely that these changes were caused by TA-GvHD, as infiltration of human CD3 or CD45-positive cells was not observed in all mice injected with $10^4$ or fewer cells (Fig 4). As it has been suggested that the development of GvHD in immunodeficient mice is dependent on the engraftment of human T lymphocytes[22, 23], our results suggest that the number of T lymphocytes transfused correlates of the risk and severity of TA-GvHD. It is well established that $10^6$ cells are required to induce chronic GvHD in NSG mice models[24], and even $10^4$ levels have demonstrated clinical GvHD[25]. Our results demonstrated that blood products with fewer than 1200–2500 mononuclear cells per blood products are below the causative levels of TA-GvHD in mice models.

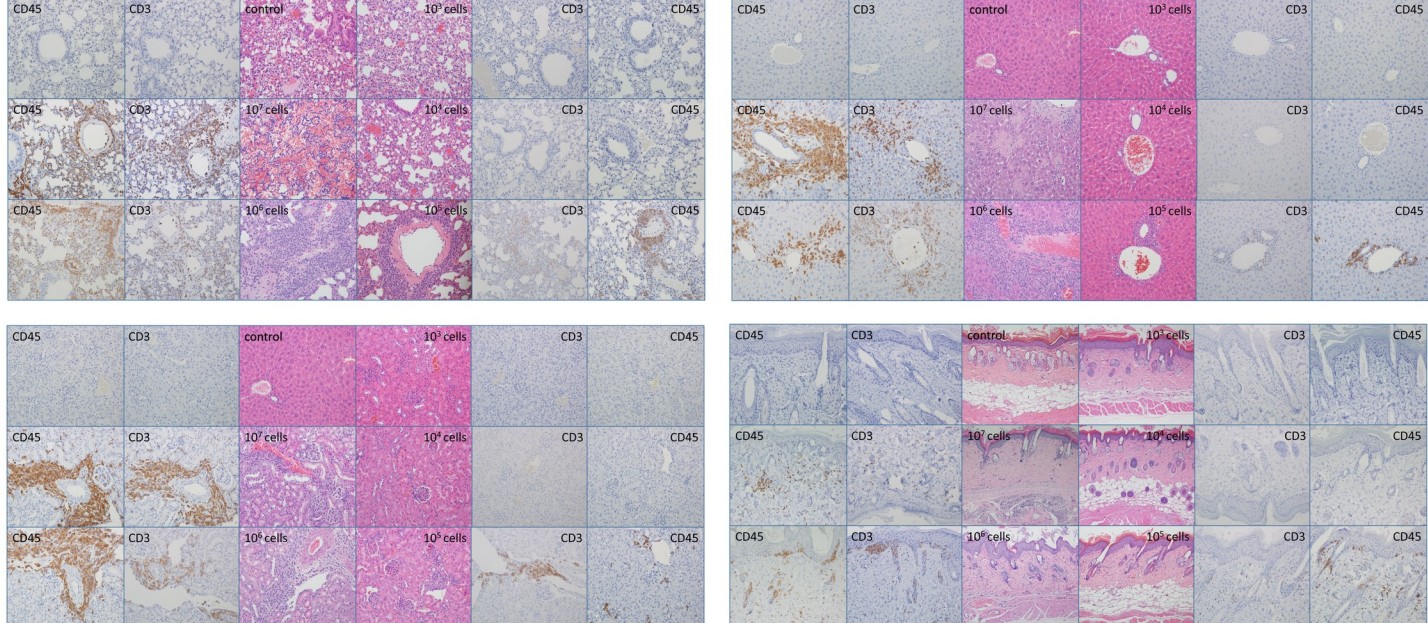

**Fig 4. Histologic findings on NOD.Cg-Prkdc<sup>scid</sup> Il2rg<sup>tm1Wjl</sup>/SzJ mice after expiration or experiment termination.** Histologic findings of the lung (upper left), liver (upper right), kidney (lower left), and skin (lower right) all show inflammation and human lymphocyte infiltration in mice injected with $10^5$ or more cells. Positive findings for graft-versus-host disease was not observed in mice injected with $10^4$ or fewer cells. All quadrants show control mice samples on the upper left, with samples from mice injected with $10^3$, $10^4$, $10^5$, $10^6$ and $10^7$ cells displayed clockwise.

## Discussion

Elimination of all white blood cells should have the same effect as inactivation of the proliferation capabilities of T cells by irradiation, but experimental data to support this hypothesis have not been published yet[26]. This study attempted to provide evidence that double-filtration of blood products does not degrade its quality while removing leukocytes to levels below detection by methods commonly used in the clinical laboratory. Also, our experimental data shows that the level of residual leukocytes (PBMCs in this study) correlates with T cell capability to proliferate and expand (*in vitro* T cell assay) or induce clinical and histological evidence of TA-GvHD (*in vivo* NSG mouse experiment). Our data suggest that double-filtered leukoreduction can be a method for risk reduction of TA-GvHD.

According to the College of American Pathologists (CAP), residual leukocytes in leukocyte reduced (LR) RBCs must be below $5 \times 10^6$/unit and the leukoreduced unit must retain at least 85% of the original number of RBCs. Regarding the hemolysis rate (%) and level of plasma potassium, they were not significantly different in the double-filtered units compared to those that had undergone single filtration. These data demonstrate that double filtration does not degrade the integrity of RBCs and does not cause the release of excessive levels of potassium. The initial high plasma free Hb levels prior to filtration observed in our samples are thought to be from the characteristics of the RBCs themselves. The samples used in this study were out-of-specification blood components due to the elevated ALT levels, and thus leukocyte-reduction filtration could be done at the time of post-processing (within the research laboratory) after several days of storage. Hence, we expected these factors would adversely affect the hemolysis rates of filtered RBCs. However, parameters for hemolysis did not increase after the two

rounds of filtration, and we conclude that perhaps these units are safer from a hyperkalemia perspective than those that had undergone gamma irradiation. Despite hemolysis being reported in RBCs that had undergone leukocyte-reduction filtration[27, 28], our experiments do not show excess levels of hemolysis after filtration. In contrast, it is well known that irradiation causes increased leakage of potassium [29]. Thus, perhaps these units are safer from a hyperkalemia perspective than those that had undergone gamma irradiation. However, as for RBC loss due to filtration. 14.1% of LR RBCs were lost following the second filtration process, in addition to 11.7% RBC loss following the first filtration. This is a limitation for an adult having several units of RBCs, but would not to be a problem for infants requiring low/small volume transfusions of irradiated RBCs [30, 31]. Other options for the removal of potassium can be considered after filtration, as the washing of the blood product, or the use of a potassium adsorption filter[32]. However, the former introduces potential risk for contamination, and the latter is still affected by the fact that irradiation causes damage to red cells[33].

With the recent improvements in blood filter technology, leukoreduction has been suggested to be an alternative to irradiation for preventing TA-GvHD, despite the concept not being generally accepted[26, 34]. Furthermore, reports on TA-GvHD from filtered blood products occurred when filtration at the bedside was more predominant, and reports from blood products after the introduction of pre-storage leukoreduction cannot be found in the literature. Leukocyte removal filters are made with porous foam material or non-woven fibers (depth filtration) or woven mesh-shaped fibers (screen filtration) [35, 36]. Although the exact mechanism of leukocyte removal still requires further investigation, it has been proposed that the most important mechanism is the adhesion of leukocytes to the filtration material. The efficiency of filtration is dependant upon the area of contact of the blood component to filtration material; in the case of foam material, the overall thickness, and in the case of fiber, the diameter of the fiber. Other factors, as surface wettability roughness, charge and hydrophilicity all play as factors of leukocyte removal efficiency. What the exact technological advancements of the filter material in modern filters are and how they compare with the filters used in the earlier case reports of TA-GvHD is not clear[3, 4]. Yet the sharp decrease in the prevalence of TA-GvHD after universal leukoreduction, according to SHOT reports of the United Kingdom [37] suggests, as universal leukoreduction was done in blood centers with modern protocols, we can speculate that when filtration is done with modern filters according to the manufacturer's directions, the risk of TA-GvHD can decrease to very low levels. We speculate that a second round of filtration of previously leukoreduced blood products can further reduce the level of residual leukocytes, as this would be doubling the opportunity to remove residual leukocytes. When comparing single and double filtration, the content of leukocytes in a single filtered unit was at levels of $2.96 \times 10^6$ cells per blood unit according to a previous study done by our group[38]. This study results showed that a second leukoreduction step resulted in having an estimated residual mononuclear cell count of 1200–2500 cells per RBC unit. Despite the difference in the counting method between the two studies, we speculate the second round of filtration resulted in a further 3 log reduction and resulted in residual cells below detection levels.

Viewing the rate of reduction in the rate of leukocyte removal is interesting, and results can be contrasted to previous studies that focused on the level of inactivated cells after irradiation. Our studies indicate that the first round of filtration results in a 5 log reduction, and the following round is indicative of a further 3 log reduction. In contrast, gamma irradiation showed that the inactivation level of T cells resulted in a range of 3–6 log reduction [7]. Pathogen inactivation methods provided better rate of inactivation, being 6.2 log for amotosalen/UVA [39], 5.9 log for amustaline/GSH [7] or 4.7–6.5 for riboflavin/UV [20, 40]. A direct comparison should not be made, as the method relied in this study was done by cell counting methods,

while the other studies were mostly done with limitation dilution assays. Furthermore, leukoreduced blood products would be unfit for limitation dilution assays, and thus direct comparison would not be feasible in future studies. However, it does suggest that the level of cell removal after the second round of filtration can result in levels of cell reduction that is substantial enough to sufficiently disable expansion.

Merely adding any filtration process may not be sufficient to replicate this result, as a previous study showing the presence of 0.06 WBCs/μL after double filtration as measured with the same method used in the present study[41]. The reason for this discrepancy is not clear, but different filters were used in the two different experiments, which might account for this. Also, as the current experimental methods were not able to determine that complete removal of all leukocytes was accomplished; a more sensitive method would be required to verify that double-filtered RBC units contain subclinical levels of residual WBCs. Other filtration methods could also be put into consideration, such as prior reduction filters [42], as a secondary filtration process. These would have to be put into strict quality assurance measures, as the number of residual WBCs would have to be warranted to certain levels with every blood product.

TA-GvHD is caused by the proliferation of T cells transfused with blood products[43]. Our *in vitro* experiments were conducted to investigate the number of mononuclear cells required for their proliferation and expansion. It can be assumed that evidence of expansion can be found when leukocyte numbers between $10^2$ and $10^3$ cells are planted in proximity, and sufficient expansion when more than $10^3$ cells are in proximity. However, any conclusion on the exact amount of T cells required for proliferation cannot be made, as *in vitro* experimentation and actual clinical conditions may differ. We can only conclude that the level of leukoreduction would have to be very efficient to completely eliminate the possibility of T cell proliferation and expansion. Further *in vivo* experimentation was done for further investigation. Mice are the primary model animals for the investigation of preclinical *in vivo* studies on GvHD [44]. Our results suggest that the number of T lymphocytes transfused correlates with the risk and severity of TA-GvHD. Our result demonstrated that infused PBMCs required for induction of GvHD was between $10^4$ and $10^5$ cells, and other studies demonstrated that transfused cell numbers as low as $10^4$ can induce clinical GvHD[25]. As we have shown that double-filtered blood products have fewer than 1200–2500 mononuclear cells per blood product, we speculate that this is below the causative levels of TA-GvHD in mice models. We suggest this as supporting evidence that further leukoreduction can significantly reduce the risk of TA-GvHD.

The conclusion of this study has some limitations. The use of irradiators is the standard in the prevention of TA-GvHD, and this study results cannot be by itself sufficient evidence to fully change clinical practice. However, the method suggested in this study is simple, and can be easily applied in the clinical environment. As the most severe adverse effect of an irradiated blood product is hyperkalemia, we suggest the use of double filtration in situations where blood irradiation is not available, or when it is not desirable. Another limitation of double filtration is that the yield of RBC recovery is inferior compared to a single filtered product (Table 2). This would mostly limit its use to low body weight patients, such as neonates and young children, as these patients would be most affected by high dosage of potassium in blood products, and would not require much amount of RBC for treatment of their anemic condition. Broad introduction of pathogen reduction techniques may further render the conclusion of this study obsolete, yet

This study's method also has limitations. Quantifying, or detecting white cells below a certain level is hindered by the detection technology itself. We have sought methods to improve this by Ficoll separation, which has its own limitations. Obtaining white cells from the buffy coat was also tried, yet failed due to the poor viability of white cells, which can be contributed

by the fact that the double filtration process requires blood products to be stored several days. Using automated analyzers in the body fluid mode was selected as a compromise, as the level of detection was acceptable for our experimentation. Further studies with more precise methods would be interesting if double-filtered blood can indeed have zero residual leukocytes within.

As the incidence of TA-GvHD is very uncommon, and irradiation is the standard of care to prevent it, this study's approach can be a starting point for testing the availability of double-filtrated blood products in selected clinical situations. In cases where irradiation is not possible, such as locations where irradiators are not available, or in cases where the risk of hyperkalemia related cardiac disorders is high (i.e., cases of kidney dysfunction, low birth weight neonates, patients on extracorporeal membrane oxygenation), double filtration might be helpful to reduce the risk of TA-GvHD if they have additional risk factors for it. The low recovery rate of RBC is a limitation, as it makes this method unfit for patients requiring multiple units of blood at once, as the use of double-filtered blood will inevitably require the use of more blood units.

## Supporting information

**S1 Checklist. NC3Rs ARRIVE Guidelines Checklist is attached in the file NC3Rs ARRIVE Guidelines Checklist.**
(PPTX)

## Author Contributions

**Conceptualization:** Jaehyun Kim, Duck Cho.

**Investigation:** Jehoon Yang, Yeup Yoon, Sangbin Han.

**Methodology:** Minh-Trang Thi Phan, Saetbyul Hong, Jungwon Kang.

**Resources:** Jehoon Yang, Jungwon Kang.

**Supervision:** Yeup Yoon, Mark H. Yazer.

**Validation:** Sangbin Han.

**Visualization:** Jaehyun Kim.

**Writing – original draft:** Sejong Chun.

**Writing – review & editing:** Mark H. Yazer, Jaehyun Kim.

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
