## [Decision Letter · Decision Letter 0]

27 Nov 2019

PONE-D-19-25999

Double-filtered leukoreduction as a method for risk reduction of transfusion-associated graft-versus-host disease

PLOS ONE

Dear Dr. Cho,

Thank you for submitting your manuscript to PLOS ONE. After careful consideration, we feel that it has merit but does not fully meet PLOS ONE’s publication criteria as it currently stands. Therefore, we invite you to submit a revised version of the manuscript that addresses the points raised during the review process.

We have now received reports from two referees of your manuscript, as agree with reviewers comments raised a few concerns about this study. After careful consideration, we invite you to submit a revised version of the manuscript.  

We would appreciate receiving your revised manuscript by Jan 11 2020 11:59PM. To enhance the reproducibility of your results, we recommend that if applicable you deposit your laboratory protocols in protocols.io, where a protocol can be assigned its own identifier (DOI) such that it can be cited independently in the future. For instructions see: http://journals.plos.org/plosone/s/submission-guidelines#loc-laboratory-protocols

We look forward to receiving your revised manuscript.

Kind regards,

Senthilnathan Palaniyandi, Ph.D

Academic Editor

PLOS ONE

Journal Requirements:

2. To comply with PLOS ONE submissions requirements, please provide methods of sacrifice in the Methods section of your manuscript.

3. As part of your revision, please complete and submit a copy of the ARRIVE Guidelines checklist, a document that aims to improve experimental reporting and reproducibility of animal studies for purposes of post-publication data analysis and reproducibility: https://www.nc3rs.org.uk/arrive-guidelines. Please include your completed checklist as a Supporting Information file. Note that if your paper is accepted for publication, this checklist will be published as part of your article.

Reviewers' comments:

Reviewer's Responses to Questions

**Comments to the Author**

1. Is the manuscript technically sound, and do the data support the conclusions?

Reviewer #1: Partly

Reviewer #2: Yes

2. Has the statistical analysis been performed appropriately and rigorously? 

Reviewer #1: No

Reviewer #2: N/A

3. Have the authors made all data underlying the findings in their manuscript fully available?

Reviewer #1: Yes

Reviewer #2: Yes

4. Is the manuscript presented in an intelligible fashion and written in standard English?

Reviewer #1: No

Reviewer #2: Yes

5. Review Comments to the Author

Reviewer #1: The authors present a study on an alternative approach (namely 2 subsequent leukoreduction by 2 distinct filtering techniques) to the current state of the art irradiation for the prevention of transfusion-associated graft-versus-host disease (TA-GVHD).

Major criticism:

1. The critical issue is to what extent does double filtration diminish the TA-GVHD triggering potential of the RBC preparation. In my opinion, this is insufficiently examined in this paper. This is attempted to be presented in the second half of Results section. About in vitro T cell expansion experiments, mere 2 sentences report. Similarly, critical circumstances are missing from the presentation of the in vivo mouse experiments of 12 lines. If one ultimately aims at an equivalency study with the gold standard technique one should have a more systematic approach to prove the hypothesis.

2. More detailed examination of quality changes following double filtration would also improve the paper. Quality parameter comparisons with those after irradiation are recommended.

3. The 2 terms: “leukocytes” and “lymphocytes” should not be used interchangeably.

4. The relationships of the tested double filtration procedure with that of the other alternative approach, namely the pathogen inactivation (reduction) should be discussed with relevant recent references (e.g. Kleinman, Transfusion 2018, Bahar, Arch Pathol Lab Med 2018). In the current text I did not find anything about this.

5. Figure legends in general must be substantially extended with additional details e.g. case numbers in the respective subgroups, rational and method of subgrouping, statistics employed etc.

6. The quality of the abstract should be improved with more numbers characterizing the experiments performed and without company names.

Reviewer #2: The authors describe a technique by which 30 units of blood where subjected twice to filtration with filters provided by the Haemonetics Corporation. With these techniques, they could no longer detect leukocytes which they calculated to be less 1200 - 2500 cells. They substantiated their findings with in-vitro studies and in-vivo with immunodeficient mice. The experimental findings appear sound, however I recommend these changes:

1) The current standard world-wide is irradiation of blood products, since TA-GVHD is universally fatal, this cannot be changed easily and this needs to be re-iterated.

2) However, inactivation procedures have been implemented for platelets in the United States. This should be discussed in detail.

3) A second filtration obviously increases the cost of blood transfusion, therefore cost saving by not using an irradiator has to be balanced against the cost of using a second filter.

6. PLOS authors have the option to publish the peer review history of their article (what does this mean?). If published, this will include your full peer review and any attached files.

Reviewer #1: No

Reviewer #2: No

---

## [Author Response · Author response to Decision Letter 0]

8 Jan 2020

Dear Editor and Reviewers.

This is the document for our response regarding the revised manuscript of the article ‘Double-filtered leukoreduction as a method for risk reduction of transfusion-associated graft-versus-host disease’. We thank you for the time dedicated to reviewing our work and hope that our revised manuscript would be more fit for publication.

Journal Requirements:

The manuscript was ensured to meed PLOS ONE’s style requirements. 

2. To comply with PLOS ONE submissions requirements, please provide methods of sacrifice in the Methods section of your manuscript.

This was added in the methods section. All mice were euthanized after experimentation, at day 43 by CO2 gas.

3. As part of your revision, please complete and submit a copy of the ARRIVE Guidelines checklist, a document that aims to improve experimental reporting and reproducibility of animal studies for purposes of post-publication data analysis and reproducibility: https://www.nc3rs.org.uk/arrive-guidelines. Please include your completed checklist as a Supporting Information file. Note that if your paper is accepted for publication, this checklist will be published as part of your article.

This was done. We have submitted a copy with our revised manuscript.

Reviewers' comments:

Comments to the Author

Reviewer #1: The authors present a study on an alternative approach (namely 2 subsequent leukoreduction by 2 distinct filtering techniques) to the current state of the art irradiation for the prevention of transfusion-associated graft-versus-host disease (TA-GVHD).

Major criticism:

1. The critical issue is to what extent does double filtration diminish the TA-GVHD triggering potential of the RBC preparation. In my opinion, this is insufficiently examined in this paper. This is attempted to be presented in the second half of Results section. About in vitro T cell expansion experiments 설명 보강, mere 2 sentences report. Similarly, critical circumstances are missing from the presentation of the in vivo mouse experiments of 12 lines. If one ultimately aims at an equivalency study with the gold standard technique one should have a more systematic approach to prove the hypothesis.

Thank you for raising this issue. We have also concerned about this in our initial manuscript. Further description of the limitation of the in vitro method, along with the meaning of our in vivo method was added. We have to admit that a systematic approach to fully prove equivalency to irradiation is impossible at the moment, and have revised our manuscript to focus on our results inidicating that a sufficient amount of cells are required to induce expansion (or GvHD, in the mouse experiment), and that our assays resulting in 1200-2500 cells per blood product is suggestive to be sufficiently below that. This itself is by no means sufficient to be an equivalency study. However, we hope that our results conclusion would shed some interest on the use of our proposed method in cases where irradiation is undesirable or impossible.

2. More detailed examination of quality changes following double filtration would also improve the paper. Quality parameter comparisons with those after irradiation are recommended.

This would be mainly summarized in Tables 1 and 2. More description was added to the manuscript on this issue. As this data would eventually be the main interest if one would apply double-filtration of RBC products, we hope that these tables would provide sufficient data to the readers.

3. The 2 terms: “leukocytes” and “lymphocytes” should not be used interchangeably.

This was also one of our concerns when submitting the initial manuscript. We have an explanation for this, and would like to clarify this issue. As leukoreduction filters target all leukocytes, in phrases where the main subject is leukoreduction itself, we used leukocytes. In the experimentation with mononuclear cells, the term PBMC or mononuclear cells were used. And when regarding the pathology of GvHD or other activities regarding T cells, the term lymphocytes were used. We have tried to be consistent on this throughout the manuscript, and did not use these as interchangeable terms. We feel that unifying these terms to a single one would be a less precise description. 

4. The relationships of the tested double filtration procedure with that of the other alternative approach, namely the pathogen inactivation (reduction) should be discussed with relevant recent references (e.g. Kleinman, Transfusion 2018, Bahar, Arch Pathol Lab Med 2018). In the current text I did not find anything about this.

Thank you for bringing up this point. It would be poor practice not to include this in a study, and we are grateful that the reviewer brought up this point. We have to admit that pathogen reduction, with the up-to-date results on this method, can be a more promising method compared to filtration. We have discussed this issue in the discussion section.

5. Figure legends in general must be substantially extended with additional details e.g. case numbers in the respective subgroups, rational and method of subgrouping, statistics employed etc.

Figure legends were revised, and to further add detail in understading the context of each figure.

6. The quality of the abstract should be improved with more numbers characterizing the experiments performed and without company names.

Abstract was revised to be more explanatory of the results of this study.

Reviewer #2: The authors describe a technique by which 30 units of blood where subjected twice to filtration with filters provided by the Haemonetics Corporation. With these techniques, they could no longer detect leukocytes which they calculated to be less 1200 - 2500 cells. They substantiated their findings with in-vitro studies and in-vivo with immunodeficient mice. The experimental findings appear sound, however I recommend these changes:

1) The current standard world-wide is irradiation of blood products, since TA-GVHD is universally fatal, this cannot be changed easily and this needs to be re-iterated.

We agree with this point. This study itself cannot be the gounds to substitute irradiation, and we suggest this as a easy method to apply when irradiation is not desirable or impossible. This was also further discussed in the discussion section. 

2) However, inactivation procedures have been implemented for platelets in the United States. This should be discussed in detail.

This was also an issue brought up by another reviewer. Thank you for commenting on this. It would be poor practice not to include this in a study, and we are grateful that the reviewer brought up this point. We have to admit that pathogen reduction, with the up-to-date results on this method, can be a more promising method compared to filtration. We have discussed this issue in the discussion section.

3) A second filtration obviously increases the cost of blood transfusion, therefore cost saving by not using an irradiator has to be balanced against the cost of using a second filter.

The cost of filtration was not a consideration when designing this study, as, said above, we do not suggest this as a substitute to irradiation, but more of an alternative method when irradiation is not desirable or not possible. This was further discussed in the discussion section. If double-filtration somehow gets further attention in the transfusion society, cost-productiveness would eventually have to be evaluated; however, as clinical trials on the efficacy of this would not be possible (due to too low prevalence and ethical issues). We feel that suggesting this as a method of last resort when irradiation cannot be done would be more fit.

---

## [Decision Letter · Decision Letter 1]

13 Feb 2020

Double-filtered leukoreduction as a method for risk reduction of transfusion-associated graft-versus-host disease

PONE-D-19-25999R1

Dear Dr. Cho,

We are pleased to inform you that your manuscript has been judged scientifically suitable for publication and will be formally accepted for publication once it complies with all outstanding technical requirements.

With kind regards,

Senthilnathan Palaniyandi, Ph.D

Academic Editor

PLOS ONE

Additional Editor Comments (optional):

Reviewers' comments:

Reviewer's Responses to Questions

**Comments to the Author**

1. If the authors have adequately addressed your comments raised in a previous round of review and you feel that this manuscript is now acceptable for publication, you may indicate that here to bypass the “Comments to the Author” section, enter your conflict of interest statement in the “Confidential to Editor” section, and submit your "Accept" recommendation.

Reviewer #1: All comments have been addressed

Reviewer #2: All comments have been addressed

2. Is the manuscript technically sound, and do the data support the conclusions?

Reviewer #1: Yes

Reviewer #2: Yes

3. Has the statistical analysis been performed appropriately and rigorously? 

Reviewer #1: Yes

Reviewer #2: Yes

4. Have the authors made all data underlying the findings in their manuscript fully available?

Reviewer #1: Yes

Reviewer #2: Yes

5. Is the manuscript presented in an intelligible fashion and written in standard English?

Reviewer #1: Yes

Reviewer #2: Yes

6. Review Comments to the Author

Reviewer #1: The criticism was adequately handled, the modifications and additions are fully acceptable. The reference is also sufficiently updated.

Reviewer #2: The comments made by the reviewers were addressed, therefore I recommend publication.

PS there are still a few typos or unusual terms, please edit carefully; the term "cells were planted" is unusual, flowers or potatoes are planted, as far lymphocytes or cells are concerned, I would recommend "cultured" or "seeded" or "incubated"

7. PLOS authors have the option to publish the peer review history of their article (what does this mean?). If published, this will include your full peer review and any attached files.

Reviewer #1: No

Reviewer #2: No

---

## [Editor Report · Acceptance letter]

11 Mar 2020

PONE-D-19-25999R1 

Double-filtered leukoreduction as a method for risk reduction of transfusion-associated graft-versus-host disease 

Dear Dr. Cho:

I am pleased to inform you that your manuscript has been deemed suitable for publication in PLOS ONE. Congratulations! Your manuscript is now with our production department. 

With kind regards,

on behalf of

Dr. Senthilnathan Palaniyandi 

Academic Editor

PLOS ONE